# G-Censor: Graph Contrastive Learning with Task-oriented Counterfactual Views

## Abstract

Graph Contrastive learning (GCL) has achieved great success in learning representations from unlabeled graph-structure data. However, how to automatically obtain the optimal contrastive views w.r.t specific downstream tasks is little studied. Theoretically, a downstream task can be causally correlated to particular substructures in graphs. The existing GCL methods may fail to enhance model performance on a given task when the task-related semantics are incomplete/preserved in the positive/negative views. To address this problem, we propose G-Censor, i.e., Graph Contrastive lEarniNg with taSk-oriented cOunteRfactual views, a model-agnostic framework designed for node property prediction tasks. G-Censor can simultaneously generate the optimal task-oriented counterfactual positive/negative views for raw ego-graphs and train graph neural networks (GNNs) with a contrastive objective between the raw ego-graphs and their corresponding counterfactual views. Extensive experiments on eight real-world datasets demonstrate that G-Censor can consistently outperform existing state-of-the-art GCL methods to improve the task performance and generalizability of a series of typical GNNs. To the best of our knowledge, this is a pioneer investigation to explore task-oriented graph contrastive learning from a counterfactual perspective in node property prediction tasks. We will release the source code after the review process.

## 1 Introduction

Inspired by the convincing success of contrastive learning in the domain of computer vision (Chen et al., 2020; He et al., 2020) and natural language processing (Gao et al., 2021), graph contrastive learning (GCL) has become an emerging field that extends the idea to graph data (You et al., 2020a; Hassani & Ahmadi, 2020; Zhu et al., 2021; Li et al., 2022), leading to generalizable, transferable and robust representations from unlabeled graph data (You et al., 2021).

Nevertheless, the generation mechanism of contrastive views, which has been recognized as an essential component in GCL (Zhu et al., 2021; Yin et al., 2022; You et al., 2021), is still facing the following challenges: **(a) Independent of downstream tasks.** Although GCL is originally proposed for self-supervised learning, how to obtain the optimal positive view when downstream tasks are available can be an important question Xie et al. (2022). However, most prior works, whether based on graph diffusion (Hassani & Ahmadi, 2020), uniform sampling (Zhu et al., 2020), or adaptive sampling (Zhu et al., 2021; You et al., 2021), ignore the downstream tasks' information. As shown in Figure 1, whether a generated view is a appropriate positive view depends critically on the downstream tasks Chen et al. (2020). **(b) Fitting spurious correlations.** To introduce task information, learnable data augmentation has been investigated to automatically obtain the positive views for downstream tasks (Yin et al., 2022). While these techniques have achieved promising performance, they are prone to be plagued by spurious correlations between graph structures and downstream tasks like general supervised methods, thus hurting the generalizability of representation model. **(c) Difficulty in negative views selection.** Beside positive views, negative sampling is also a vital component in GCL. Contrastive learning can benefit from hard negative samples (Joshua et al., 2021). Meanwhile, negative samples, actually similar to the raw instances, can lead to a performance drop (Chuang et al., 2020). Therefore, it can be hard to select suitable negative samples. Some works (He et al., 2020) utilize a great number of negative samples to avoid this trade-off but

may cause scalability problems. These challenges can become more non-trivial with graph data since graph data are far more complex due to the non-Euclidean property (Zhu et al., 2021).

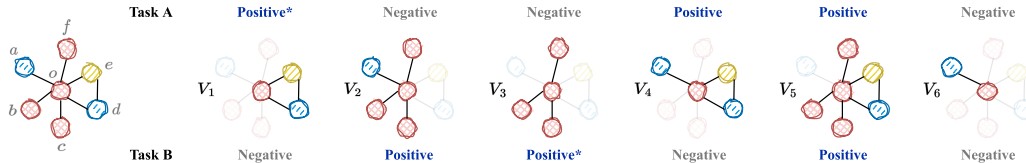

Figure 1: An illustration for task-oriented contrastive views. Task A is to predict whether a node is in a triangle and task B is to predict the color of a node. A task-oriented view is positive if and only if it contains the credible evidence for the task label, otherwise it should be negative.

In this paper, we propose a novel model-agnostic framework for node property prediction tasks, namely G-CENSOR, i.e., Graph Contrastive lEarniNg with taSk-oriented cOunteRfactual views. G-CENSOR generates high-quality positive and negative views simultaneously from a counterfactual perspective. In other words, a task-oriented counterfactual question about the contrastive views could be asked: *"Would a judgment on the task label of an ego-graph change if part of the structure of the ego-graph were erased?"* The answer *no* should be assigned to a positive view while the answer to a negative view should be *yes*. Technically, G-CENSOR adopts the learnable view generation approach and leverages an original counterfactual optimization objective to decompose an ego-graph into the sub-structures causally correlated to the downstream tasks and the sub-structures spuriously correlated to the downstream tasks. This two parts can further be regarded as positive and negative views, respectively. Learning representation model with such contrastive views can enhance both the model's task performance and generalizability. Notably, G-CENSOR doesn't need to contrast in-batch negative samples, this characteristic can help G-CENSOR get rid of performance-memory trade-off inherent in most prior GCL methods.

The core contribution of this paper can be three-fold: **(a)** To the best of our knowledge, this is a pioneer investigation to explore task-oriented graph contrastive learning from a counterfactual perspective in node property prediction tasks. **(b)** We develop a novel model-agnostic framework, G-CENSOR, an approach to automatically generate both task-oriented counterfactual positive and negative views to enable sufficient and efficient graph contrastive learning. **(c)** We conduct extensive experiments on eight real-world datasets to demonstrate the superiority of G-CENSOR over existing state-of-the-art GCL methods to improve the performance and generalizability of typical GNNs.

## 2 RELATED WORKS

### 2.1 GRAPH CONTRASTIVE LEARNING

Inspired by the success of data augmentation and contrastive learning in texts and images to address the data noise and data scarcity issues, Many graph contrastive learning (GCL) frameworks have been proposed lately (Liu et al., 2022; Xie et al., 2022). GCL works usually consist of a graph views generation component to construct positive and negative views, and a contrastive objective to discriminate positive pairs from negative pairs (Xie et al., 2022). Most works generate positive views by uniform random transformation, e.g. node dropping, edge perturbation and subgraph sampling (Zhu et al., 2020; You et al., 2020b; Yu et al., 2020; Zhao et al., 2021; You et al., 2020a; Hassani & Ahmadi, 2020; Sun et al., 2020; Velickovic et al., 2019). Zhu et al. (2021) proposed the adaptive data augmentation strategies to reflect input graph's intrinsic patterns, i.e., assign larger drop probabilities to unimportant edges. Recently, several works have proposed trainable augmentation strategies (You et al., 2021; Li et al., 2022; Yin et al., 2022) to learn drop probability distribution over nodes or edges. However, few works have discussed on how to generate an optimal task-oriented positive and negative views for graph data, and no learnable augmentation strategy has been proposed for node property prediction task. Table 1 lists the comparison between G-CENSOR and the other state-of-the-art GCL models on 4 different properties.

Table 1: The comparison between G-CENSOR and the other state-of-the-art GCL models.

| Property | MVGRL | GRACE | GCA | BGRL | AutoGCL | RGCL | **G-CENSOR** |
|---|---|---|---|---|---|---|---|
| Primary task | node&graph | node | node | node | graph | graph | **node** |
| Task-oriented | no | no | no | no | yes | no | **yes** |
| Positive views | uniform | uniform | adaptive | uniform | learnable | learnable | **learnable** |
| Negative views | in-batch | in-batch | in-batch | - | in-batch | learnable&in-batch | **learnable** |

## 2.2 COUNTERFACTUAL GRAPH EXPLANATION

Recently, the research on explainability of GNNs is experiencing a rapid development (Yuan et al., 2021). Most of GNN explanation methods (Ying et al., 2019; Luo et al., 2020) focus on identifying a subgraph of the original graph that contributes most to the prediction of a trained GNN. There are also works make explanations based on retrieving similar existed instances (Faber et al., 2020). Since removing the explanation subgraph from the input graph does not necessarily change the prediction(Bajaj et al., 2021), some counterfactual GNN explanation techniques (Lucic et al., 2022; Bajaj et al., 2021) have been proposed. These techniques address a problem that identifies a small subset of edges of the input graph instance such that removing those edges significantly result in an alternative prediction (Lucic et al., 2019; 2022; Bajaj et al., 2021). While all above works deal with post hoc explanations of estimated GNNs, this study emphasizes ad hoc detection of the causally correlated subgraph with respect to tasks. See Section 4.2 for details.

## 3 PRELIMINARIES AND PROBLEM FORMULATION

In this section, we provide formal definitions of the optimal task-oriented counterfactual positive/negative views with insightful explanations, and formulate the research problems.

Let $\mathcal{G} = (\mathcal{V}, \mathcal{E})$ be an undirected graph with nodes $v_i \in \mathcal{V}$ and edges $(v_i, v_j) \in \mathcal{E}$. For each node $v_i$, a feature description $\boldsymbol{x}_{v_i} = [x_1, x_2, \cdots, x_i, \cdots, x_d]$ and a task label $y_{v_i} \in \{1, \ldots, c\}$ are assigned, where $d$ is feature dimension and $c$ is the number of classes. A $k$-hop ego-graph (Daly & Haahr, 2007) of a node $v_i$ can be defined as $G_{v_i,k} = (V_{v_i,k}, E_{v_i,k}, v_i)$, where $v_i$ indicates the ego, $V_{v_i,k} \subseteq \mathcal{V}$ is the set of all the nodes that are at most $k$ hops away from $v_i$, and $E_{v_i,k} \subseteq \mathcal{E}$ is the set of interconnected edges between nodes in $V_{v_i,k}$. $G'_{v_i,k} = (V'_{v_i,k}, E'_{v_i,k}, v_i)$ can be an ego-subgraph of $G_{v_i,k}$, where $E'_{v_i,k} \subseteq E_{v_i,k}$, and $V'_{v_i,k}$ is nodes involved in $E'_{v_i,k}$. The complement of $G'_{v_i,k}$ w.r.t $G_{v_i,k}$ is defined by $\bar{G}'_{v_i,k} = (\bar{V}'_{v_i,k}, \bar{E}'_{v_i,k}, v_i)$, where $\bar{E}'_{v_i,k}$ is $E_{v_i,k} \backslash E'_{v_i,k}$, and $\bar{V}'_{v_i,k}$ is the nodes involved in $\bar{E}'_{v_i,k}$. Note that $G^+_{v_i,k}$ and $G^-_{v_i,k}$ must be connected subgraphs thus isolated edges in them will be removed. For brevity, we will omit $v_i$ and $k$ in the subsequent sections.

**Definition 1** (Optimal Task-oriented Counterfactual Positive View $G^+$). *An ego-subgrah $G'$ is the optimal task-oriented counterfactual positive view $G^+$ for an ego-graph instance $G$ if and only if $E'$ contains and only contains all the edges that are causally correlated to the task label.*

**Definition 2** (Optimal Task-oriented Counterfactual Negative View $G^-$). *An ego-subgrah $G'$ is the optimal task-oriented counterfactual negative view $G^-$ for an ego-graph instance $G$ if and only if $\bar{G}'$ is the complement of the optimal task-oriented counterfactual positive view $G^+$ w.r.t $G$.*

Figure 2 explains $G^+$ and $G^-$ from a perspective of ego-graph generation process for node property prediction tasks. In both cases, the relationship between $E^1$ and y can be stable since $E^1$ is the direct cause or effect of y, i.e., causally-correlated to y. The joint distribution of $E^0$ with y would be different if ego set changes, since $E^0$ and y are spuriously correlated(Schölkopf et al., 2012; Joshi & He, 2022) conditioned on ego, i.e.v. Therefore, according to the Definition 1, $G^+$ should be $G^1 = (V^1, E^1, v)$; according to the Definition 2, $G^-$ should be $G^0 = (V^0, E^0, v)$.

Now let $\phi(G) : \{G \mid v_i \in \mathcal{V}\} \rightarrow \mathbb{R}^c$ be any GNN that maps an ego-graph $G$ to a probability distribution $\boldsymbol{p}$ of the ego over the label space $\mathbb{R}^c$. Contrastive learning with task-oriented counterfactual views is defined as:

**Definition 3** (Graph Contrastive Learning with Task-oriented Counterfactual Views). *Given a $G$ with its $G^+$ and $G^-$, learning a function $\phi$ that maximizes the consistency between pair $(\phi(G), \phi(G^+))$ compared with pair $(\phi(G), \phi(G^-))$.*

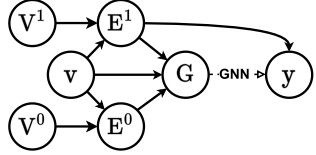
(a) Graph structure determines label.

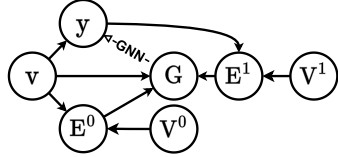
(b) Label of ego causes graph structure.

Figure 2: Causal diagrams (Pearl, 2009) for the 1-hop ego-graph generation process. A solid arrow from a to b represents a causal relationship from a to b. The two dashed lines with text of GNN represent that the whole observed graph (G) is used to predict task label (y). **(a)** The attributes of the ego (v) can cause two kinds of edges: $E^0$ (with nodes $V^0$) and $E^1$ (with nodes $V^1$). Only the structure of ($E^1$) determines the label (y). **(b)** The label (y) is an inherited property of the ego (v) and this property causes $E^1$ (with nodes $V^1$). Meanwhile, other properties of v would cause ($E^0$). For interpretative examples, refer to Appendix A.1; for the causal diagrams of $k$-hop ($k > 1$) ego-graph generation process, refer to Appedix A.2.

As Figure 2 shows, GNNs trained in a normal flow may fit the spurious correlations between $G^-$ and y since they take the whole observed graph G as input. Learning such non-causal association would reduce the reliability and the generalizability of the GNNs. In this study, we train GNNs with the auxiliary graph contrasting learning with task-oriented counterfactual views. Our goal is to make the learned representation to be as independent as possible from $G^-$, so that the model can achieve better generalizability.

Assuming that we can have a $\phi$ that is a sufficient encoder (Chen et al., 2020), where an input graph $G$ can be encoded without information loss. Let $R(\phi)$ represents the empirical risk of estimation and $L(\phi(G), y)$ represents the estimation loss, we can make a following assumption:

**Assumption 1.** *Given a set $\{(G,\ y)\}$, $\phi$ estimated with $\{(G^-,\ y)\}$ suffers a greater empirical risk compared to $\phi$ estimated with $\{(G^+,\ y)\}$ if $\phi$ is sufficient, i.e., $R(\hat{\phi}^-) = \mathbb{E}[L(\hat{\phi}^-(G^-), y)] > R(\hat{\phi}^+) = \mathbb{E}[L(\hat{\phi}^+(G^+), y)]$, where $\hat{\phi} = \underset{\phi}{\arg\min} R(\phi)$ and $L(\phi(G), y)$ can be $\mathbf{1}_{\underset{i}{\arg\max}[\phi(G)]_i \neq y}$.*

In Appedix A.1, we provide an example to explain the reasonableness of Assumption 1. Let $\varphi(G; \Lambda)$ be a function parameterized by $\Lambda$, which takes $G$ as input and produces the optimal task-oriented positive/negative views. We can enforce $\varphi$ satisfy the signature in Assumption 1. Based on the above definitions and assumption, in this paper, we investigate the following two research questions:

**Question 1.** How to design and learn a model-agnostic $\varphi$ under the Assumption 1 to conduct graph contrastive learning with task-oriented counterfactual views?

**Question 2.** Can we really enhance $\phi$ on given tasks with graph contrastive learning with task-oriented counterfactual views produced by $\varphi$?

## 4 METHODOLOGY

**To answers the Question 1**, we illustrate the proposed framework G-CENSOR in this section. The overall architecture of G-CENSOR is shown in Figure 3.

### 4.1 BASE GNN MODELS

We select GraphSAGE Hamilton et al. (2017) (abbreviated as SAGE), GAT Veličković et al. (2018) and GIN Xu et al. (2019) as base GNN models, i.e., $\phi$, to validate the G-CENSOR. These three models were chosen because they are representative and have been applied in many real scenarios. For details of these three models, refer to Appendix B.

Then we can minimize a general prediction loss $\mathcal{L}_{pred}$, e.g., the negative log likelihood (NLL), to get the estimated model $\hat{\phi}$ as follows:

$$\mathcal{L}_{pred}(\phi) = \mathbb{E}_{\{G_v | \forall v \in \mathcal{V}\}} \left[ \text{NLL}(\phi(G_v; \Theta), y_v) \right] \tag{1}$$

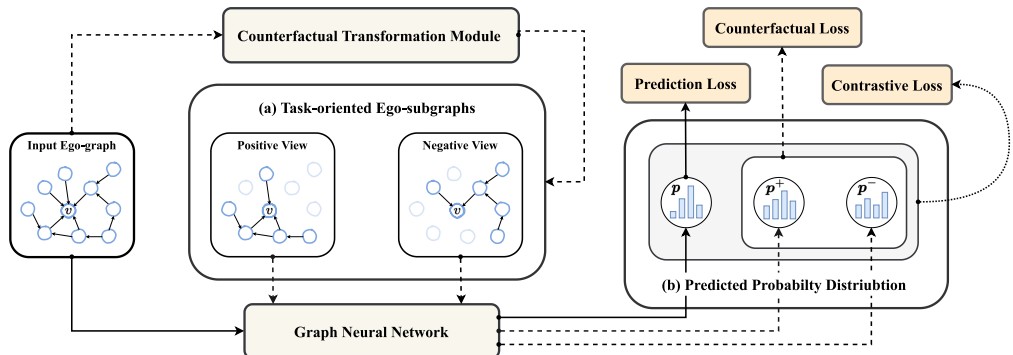

Figure 3: The overall architecture of the G-CENSOR. The solid lines represent a normal flow to train a GNN $\phi$ with the raw ego-graphs by minimizing a general prediction loss (Section 4.1). The dashed lines represent the flow to train a counterfactual transformation module $\varphi$ that generates task-oriented counterfactual positive/negative views by minimizing a designed counterfactual loss (Section 4.2). The dotted line is conducting graph contrastive learning with task-oriented counterfactual views (Section 4.3). We jointly optimize the three objectives to learn $\phi$ and $\varphi$ (Section 4.4).

## 4.2 COUNTERFACTUAL TRANSFORMATION MODULE (CTM)

Figure 4 depicts the proposed counterfactual transformation module for ego-graph, i.e., $\varphi$, to estimate the probability that an edge is part of the optimal task-oriented counterfactual positive view $G^+$ for a $G$.

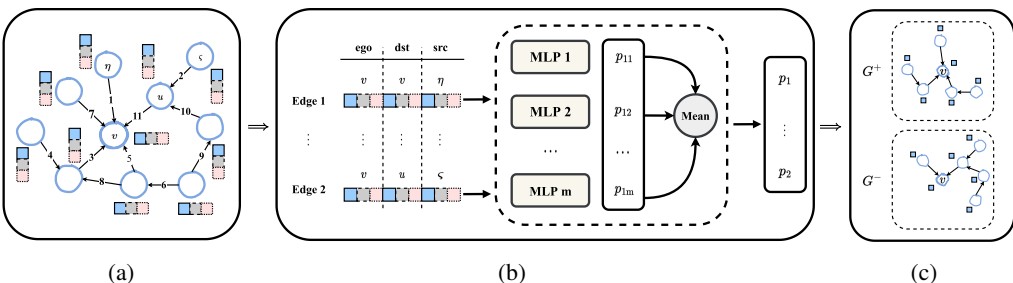

Figure 4: Counterfactual Transformation Module. (a) Solid blue square is original node feature $\boldsymbol{x}$, dashed gray square is landing probability of random walk from the ego to a node, i.e., $\boldsymbol{r}$, and dotted pink square is whether a node is the ego, i.e., $\mathbf{1}_{u=v}$. (b) The estimation of the probability that an edge belongs to $G^+$. (c) Sampling $G^+$ according to the estimated probability of all the edges and constructing $G^-$ based on $G^+$. Only original node feature $\boldsymbol{x}$ keeps in both $G^+$ and $G^-$.

**Representation of an edge.** To make an edge as distinguishable as possible, we preserve both the feature and structural information in the representation of an edge. Technically, for a node $u$ in an ego-graph $G_v$, we first assign an ego identifier defined as $\mathbf{1}_{u=v}$ and then construct auxiliary structural information leveraging Distance Encoding (DE) Li et al. (2020) defined as landing probabilities of random walks of different lengths from ego $v$ to the node $u$, which is denoted by $\boldsymbol{r}_u$. Therefore, a new node feature vector for $u$ is given as

$$\tilde{\boldsymbol{x}}_{u|G_v} = \boldsymbol{x}_u \parallel \boldsymbol{r}_{u|G_v} \parallel \mathbf{1}_{u=v}, \text{ where } \boldsymbol{r}_{u|G_v} = [(\boldsymbol{A}\boldsymbol{D}^{-1})_{u,v}, (\boldsymbol{A}\boldsymbol{D}^{-1})^2_{u,v}, \dots, (\boldsymbol{A}\boldsymbol{D}^{-1})^l_{u,v}]. \quad (2)$$

$\boldsymbol{A}$ is the adjacency matrix for $G_v$ such that $\boldsymbol{A}_{u,\varsigma} = 1$ iff $(u,\varsigma) \in E_v$ and $\boldsymbol{D}$ is the degree (diagonal) matrix for $G_v$ where $\boldsymbol{D}_{u,u}$ is the degree of node $u$. $l$ indicates the length of random walks, which is set to the same as $k$. Then an edge $(u,\varsigma)$ in $G_v$ is represented as

$$\boldsymbol{e}_{u\varsigma|G_v} = \tilde{\boldsymbol{x}}_{v|G_v} \parallel \tilde{\boldsymbol{x}}_{u|G_v} \parallel \tilde{\boldsymbol{x}}_{\varsigma|G_v}. \quad (3)$$

**Probability that an edge belongs to $G_v^+$.** For simplicity, we assume the binary variable $\mathbf{1}_{(u,\varsigma) \in E_v^+}$ follows a Bernoulli distribution, i.e., $\mathbf{1}_{(u,\varsigma) \in E_v^+} \sim \mathrm{Ber}(\theta_{u\varsigma|G_v})$ and $\theta_{u\varsigma|G_v}$ is estimated by multiple multilayer perceptrons (MLPs) as follows:

$$P\left((u,\varsigma) \in E_v^+\right) = \theta_{u\varsigma|G_v} = \mathrm{S}\left(\frac{1}{m}\sum_{i=1}^m \mathrm{MLP}_i\left(\boldsymbol{e}_{u\varsigma|G_v}; \Lambda_i\right)\right), \qquad (4)$$

where $\mathrm{S} = \frac{e^x}{1+e^x}$ is the sigmoid funtion.

Assuming all the edges are independent of each other, the probability of a sub-egograph $G_v^{'}$ being $G_v^+$, i.e., $P(G_v^+ = G_v^{'})$ can be obtained by

$$P(G_v^+ = G_v^{'}) = \Pi_{(u,\varsigma)\in E_v^{'}} P\left((u,\varsigma) \in E_v^+\right) \Pi_{(u,\varsigma)\in E_v \setminus E_v^{'}} \left(1 - P\left((u,\varsigma) \in E_v^+\right)\right). \qquad (5)$$

Note that once $G_v^+$ has been obtained, we can further construct $G_v^- = \bar{G}_v^+$. Due to the discrete nature, we adopt the reparameterization trick Jang et al. (2017); Maddison et al. (2017) to enable updating parameters in the MLPs with general gradient-based optimizer. See Appendix C for details.

**Computational Complexity.** The complexity for the edge representation $\boldsymbol{e}_{u\varsigma|G_v}$ is $O(l \cdot |V_v| \cdot |E_v|)$ due to the computation of $\tilde{\boldsymbol{x}}_{u|G_v}$ (Yuster & Zwick, 2005) and the complexity for the probability estimation is $O((1 + \frac{h-1}{m}) \cdot |E_v| \cdot d^2)$, where we assume the hidden dimension in $\mathrm{MLP}_i$ is equal to $\frac{d}{m}$ and $h$ is the number of layers in $\mathrm{MLP}_i$. As a comparison, the complexity for GNN-based flow used in learnable views generation (Yin et al., 2022; Li et al., 2022) is $O(k \cdot |E_v| \cdot C_e + k \cdot |V_v| \cdot C_v + h \cdot |E_v| \cdot d^2)$ where $C_e$ and $C_v$ are the complexity of message passing and combination respectively. In practice, the latter is likely to be greater than the former due to large $C_e$ and $C_v$. See Appendix D for the real runtime evaluation.

**Optimization objective.** To satisfy the signature of $G_v^+$ in Definition 1, the minimum NLL loss on $\phi^+(G_v^+; \Theta^+)$ can be minimized to encourage $G_v^+$ sufficient to predict the correct label $y_v$ as follows:

$$\mathcal{L}_{suff}(\varphi \mid \hat{\phi}^+) = \mathbb{E}_{G_v^+ \sim P(G_v^+ = G_v^{'})}\left[\mathcal{L}_{pred}(\hat{\phi}^+)\right], \qquad (6)$$
$$\text{where } \hat{\phi}^+ = \underset{\phi^+}{\arg\min}\mathcal{L}_{pred}(\phi^+).$$

Under Assumption 1, the relationship between $G_v^+$ and $G_v^-$ can be established by a marginal rank loss that enforces $\hat{\phi}^-(G_v^-; \hat{\Theta}^-)$ results in a greater empirical risk.

$$\mathcal{L}_{rank}(\varphi \mid \hat{\phi}^+, \hat{\phi}^-) = \mathbb{E}_{G_v^+ \sim P(G_v^+ = G_v^{'})}\left[\max\left(0, -\mathcal{L}_{pred}(\hat{\phi}^-) + \mathcal{L}_{pred}(\hat{\phi}^+) + \delta_{margin}\right)\right], \quad (7)$$
$$\text{where } \hat{\phi}^- = \underset{\phi^-}{\arg\min}\mathcal{L}_{neg}(\phi^-).$$

Additionally, to enforce $\varphi$ to find the most significant edges, $G_v^+$ should be as sparse as possible, which aligns with the findings of previous studies in causal discovery and counterfactual explanation (Zheng et al., 2018; Wachter et al., 2017).A L1 regularization $\mathcal{L}_{size}$ is applied on the probability of being selected into $E_v^+$ for all edges:

$$\mathcal{L}_{size}(\varphi) = \mathbb{E}_{\forall(u,\varsigma)\in E_v^+}\left[P\left((u,\varsigma) \in E_v^+\right)\right]. \qquad (8)$$

Note that the prior proportion of the causally-correlated part is different in different domains. With a parameter $\alpha$ to control $\mathcal{L}_{size}(\varphi)$, the counterfactual loss for the counterfactual transformation module can be formulated as:

$$\mathcal{L}_{cf}(\varphi) = \mathcal{L}_{suff}(\varphi \mid \hat{\phi}^+) + \mathcal{L}_{rank}(\varphi \mid \hat{\phi}^+, \hat{\phi}^-) + \alpha \cdot \mathcal{L}_{size}(\varphi). \qquad (9)$$

**The Difference with Counterfactual Explanation**. To address task rather than model, $\mathcal{L}_{suff}$ maximizes the mutual information between $G_v^+$ and $y_v$ while counterfactual explanation works maximize the mutual information between $G_v^+$ and $\hat{\phi}(G_v; \Theta)$. Counterfactual explanation works directly maximize the empirical risk $\mathcal{L}_{pred}(\phi^-)$, which is helpful to explain an estimated model but can sort spurious structures into $G_v^+$ since spurious structures is also correlated to label. Therefore, we choose a rank loss to ensure consistency with Assumption 1.

### 4.3 CONTRASTIVE LEARNING WITH TASK-ORIENTED COUNTERFACTUAL VIEWS

Contrasting the raw ego-graphs to both the positive and negative counterfactual views can inject task knowledge from $\varphi$ into $\phi$ and lead to better model performance and generalizability. An InfoNCE (van den Oord et al., 2018) style loss is formulated as

$$\mathcal{L}_{cl}(\phi) = \mathbb{E}_{\{(G_v, G_v^+, G_v^-) | \forall v \in \mathcal{V}\}} - \log \left( \frac{\exp\left(\text{sim}(G_v, G_v^+)/\tau\right)}{\exp(\text{sim}(G_v, G_v^+)/\tau) + \exp\left(\text{sim}(G_v, G_v^-)/\tau\right)} \right), \quad (10)$$

where $\text{sim}(G_v, G_v') = 1 - \text{JSD}(\phi(G_v), \phi(G_v'))$ and JSD is the Jensen–Shannon divergence. $\tau$ is the temperature parameter and plays a role in controlling the strength of penalties on the task-oriented counterfactual negative views. The larger the $\tau$ is, the smaller the influence of similarity between original ego-graphs and the task-oriented counterfactual negative views is.

**Scalability**. Similar to BGRL (Thakoor et al., 2021), this contrastive loss doesn't need to contrast in-batch negative samples and has time and space complexities scaling linearly to the batch size. Thus, G-CENSOR can get rid of performance-memory trade-off inherent in most prior GCL methods.

### 4.4 JOINT LEARNING OF GNN AND CTM

For simplicity and efficiency, we empirically share $\phi$ with $\phi^+$ and $\phi^-$, and jointly learn $\phi$ and $\varphi$ with a weight parameter $\beta$ for the contrastive learning. Therefore, a final joint loss can be formulated as:

$$\mathcal{L}_{joint}(\phi, \varphi) = \mathcal{L}_{pred}(\phi) + \mathcal{L}_{cf}(\varphi \mid \phi) + \beta \mathcal{L}_{cl}(\phi). \quad (11)$$

## 5 EXPERIMENTS

**To answer the Question 2**, we conduct extensive experiments on eight real-world datasets from two perspectives: task performance and model generalizability.

### 5.1 EXPERIMENTAL SETUP

**Dataset.** Eight open benchmark datasets across three domains are investigated: (a) five citation networks (Bojchevski & Günnemann, 2018) including CoraFull, CoraML, CiteSeer, DBLP and PubMed. (b) two product networks (Shchur et al., 2018) including Computers and Photo. (3) an image network (Zeng et al., 2020) Flickr. See Appendix E.1 for more details of datasets.

**Baselines.** Three types of baselines are compared: (a) base models, i.e., SAGE, GAT and GIN. (b) GCL methods with uniform or adaptive data augmentation, i.e., MVGRL(Hassani & Ahmadi, 2020), GRACE(Zhu et al., 2020), GCA (Zhu et al., 2021) and BGRL (Thakoor et al., 2021). (c) GCL methods with learnable data augmentation, i.e., AUTOGCL (Yin et al., 2022) and RGCL (Li et al., 2022). For more details, refer to Appendix E.2.

**Implementation.** For a fair comparison, the architecture of base GNNs and batch training setting were the same for all methods. All the experiments were run 5 times with random seeds from 0 to 4. For more implementation details, refer to Appendix E.3.

### 5.2 IMPROVEMENT ON PERFORMANCE

As shown in Table 2, G-CENSOR can significantly enhance the performance of base GNNs (best in 23 out of 24 settings). Specifically, G-CENSOR improves the accuracy of the base GNNs from 0.4% (GraphSAGE on CiteSeer) to 20.67% (GraphSAGE on CiteSeer). Considering all settings, the average gain of G-CENSOR is around 4.6%. Morever, G-CENSOR can consistently outperform the SOTA methods from 0.06% (GAT on Computers) to 14.59% (GIN on Flickr) except for setting of GAT on Flickr, where G-CENSOR got the second best performance. The average gain of G-CENSOR against SOTA is 1.87%. Note that on datasets like Photo, though the SOTA methods have already achieved a high performance (>93%), G-CENSOR can still pushes the boundary forward (>94%). All these results can prove the ability of G-CENSOR to enhance model performance. Meanwhile, GCA, an approach that adopts adaptive data augmentation and thus enables models learn important

Table 2: Comparison in node classification accuracy on test set with independent and identical distribution. **Bold** and underline indicate the best and second performance, respectively.

| | Method | CiteSeer | Computers | Cora_Full | Cora_ML | DBLP | Flickr | Photo | PubMed |
|---|---|---|---|---|---|---|---|---|---|
| | BASE | 91.25±0.28 | 85.62±2.05 | 55.72±1.17 | 83.83±1.48 | 80.97±0.99 | 42.38±0.24 | 92.99±0.35 | 85.34±0.70 |
| | MVGRL | 65.83±1.48 | OOM | OOM | 44.36±7.12 | 74.19±0.97 | TLE | OOM | 86.16±0.71 |
| | GRACE | 89.83±0.45 | 86.67±1.40 | 56.47±0.88 | 84.13±1.57 | 81.74±0.71 | 49.30±0.56 | 93.48±1.07 | 86.86±0.39 |
| | GGA | 90.20±0.88 | 84.71±1.74 | 56.87±1.34 | 84.52±0.90 | 81.81±0.56 | 49.76±1.00 | 92.50±1.04 | 86.85±0.33 |
| | BGRL | 90.17±1.00 | 86.25±0.65 | 56.40±1.25 | 83.64±2.43 | 81.35±0.32 | 50.69±0.25 | 91.90±1.83 | 86.50±0.29 |
| SAGE | RGCL | 89.29±0.87 | 58.31±12.38 | 49.00±1.89 | 80.68±1.66 | 80.24±0.61 | 38.91±7.54 | 81.26±7.22 | 81.85±0.52 |
| | AUTOGCL | 90.26±0.39 | 85.73±0.88 | 56.30±0.72 | 83.98±0.79 | 81.33±1.21 | 40.99±1.80 | 92.40±0.93 | 86.45±0.69 |
| | G-CENSOR | **91.65±0.26** | **88.65±0.47** | **58.10±0.91** | **85.19±1.06** | **84.01±0.46** | **51.14±0.15** | **94.52±0.25** | **88.17±0.39** |
| | BASE | 90.19±0.51 | 89.60±0.38 | 58.11±1.19 | 83.56±1.39 | 81.20±0.58 | 47.31±4.31 | 92.88±0.71 | 85.25±0.70 |
| | MVGRL | 88.64±0.39 | OOM | OOM | 81.96±1.11 | 81.25±0.77 | TLE | OOM | 86.10±0.81 |
| | GRACE | 89.77±0.33 | 90.15±0.41 | 59.28±0.74 | 84.22±2.21 | 82.52±0.49 | 51.56±0.29 | 93.32±0.45 | 86.34±0.62 |
| | GGA | 90.22±0.71 | 90.56±0.45 | 59.62±0.52 | 84.49±0.73 | 82.60±0.54 | **51.91±0.69** | 93.49±0.68 | 86.75±0.27 |
| GAT | BGRL | 90.09±1.17 | 89.46±0.88 | 58.13±0.78 | 84.44±1.29 | 81.52±0.83 | 50.55±1.12 | 92.50±0.70 | 85.99±0.40 |
| | RGCL | 88.94±1.56 | 61.64±23.22 | 45.60±0.92 | 79.64±1.58 | 80.06±1.55 | 42.38±0.24 | 86.56±3.32 | 80.83±1.93 |
| | AUTOGCL | 90.27±0.87 | 89.54±0.80 | 58.69±1.33 | 82.73±0.85 | 82.14±0.66 | 49.97±0.87 | 93.16±0.65 | 86.20±0.42 |
| | G-CENSOR | **91.26±0.68** | **90.61±0.34** | **60.21±0.24** | **85.85±1.15** | **84.06±0.65** | 51.66±0.22 | **94.13±0.46** | **87.28±0.35** |
| | BASE | 89.36±0.80 | 82.97±2.20 | 56.40±0.61 | 81.84±0.96 | 79.52±0.74 | 45.31±2.56 | 88.25±1.66 | 86.05±0.41 |
| | MVGRL | 85.90±1.14 | OOM | OOM | 71.27±3.32 | 81.47±0.76 | TLE | OOM | 85.18±0.78 |
| | GRACE | 88.74±0.85 | 88.35±0.86 | 55.96±0.68 | 83.01±0.95 | 79.64±1.00 | 44.11±1.19 | 93.21±0.69 | 86.83±0.21 |
| | GGA | 89.53±0.22 | 85.96±1.69 | 56.45±0.89 | 83.77±1.03 | 80.12±0.81 | 42.76±1.18 | 92.75±1.49 | 86.80±0.51 |
| GIN | BGRL | 89.93±0.85 | 88.45±1.34 | 56.83±0.85 | 82.98±1.05 | 80.66±0.97 | 43.49±1.15 | 91.95±1.47 | 86.86±0.26 |
| | RGCL | 88.50±1.36 | 82.95±1.80 | 52.19±0.75 | 80.42±0.83 | 79.85±0.62 | 43.90±2.66 | 89.34±1.06 | 81.61±0.25 |
| | AUTOGCL | 89.17±1.11 | 88.46±1.87 | 55.21±0.89 | 82.39±1.34 | 79.91±0.55 | 37.66±1.93 | 92.40±1.87 | 86.69±0.92 |
| | G-CENSOR | **90.98±0.56** | **90.36±0.54** | **58.46±0.70** | **84.52±0.72** | **83.42±0.23** | **51.92±0.17** | **94.57±0.06** | **87.66±0.23** |

[1] OOM means Out Of Memory (>32GB) and TLE means Time Limit Exceeded (seconds per epoch > 1000s)

structures, outperforms other baselines in most settings. This may imply the effectiveness of G-CENSOR to auto-select the task-oriented positive structures. As for learnable data augmentation methods, i.e., AUTOGCL and RGCL, they didn't achieve satisfactory performance on most settings probably because of the task shift from graph classification to node classification.

## 5.3 IMPROVEMENT ON GENERALIZABILITY

To verify the ability of G-CENSOR to boost the generalizability of GNNs, we further conduct experiments on **out-of-distribution data split setting** based on the confounder discussed in 3, i.e., the ego. For each dataset, we first run a NODE2VEC (Grover & Leskovec, 2016) to get nodes' embeddings and cluster the nodes to two clusters by K-MEANS (Lloyd, 1982). The cluster with larger sample size is randomly divided into training and validation sets and the other is regarded as testing set. As shown in Table 3, G-CENSOR still significantly enhanced the performance of base GNNs up to 18.26% with an average improvement of 6.02% and consistently outperformed the SOTA methods up to 7.40% with an average improvement of 1.62%. While a larger performance degradation suggests a larger distribution gap between the test set and the training/validation set, it's observed that base GNNs can benifit from G-CENSOR compared to SOTA methods. For example, Cora_Full, Photo and Flickr are the three datasets with the most significant performance degradation but G-CENSOR outperformed the best SOTA the most on these three datasets. All these results demonstrate the superiority of G-CENSOR to enhance the generalizability of various GNNs.

## 5.4 SENSITIVITY ANALYSIS

G-CENSOR's sensitivity w.r.t. hyperparameters $\alpha$, $\delta$, $\tau$ and $\beta$ (with GAT as base model) is evaluated by presenting the median performances in Figure 5. It's observed that **(a)** $\alpha$ plays a crucial role in G-CENSOR. Different tasks prefer to different $\alpha$, e.g., the best $\alpha$ is around 0.1 on Cora but is around 0.005 on Photo. And increasing $\alpha$ constantly can potentially hurt the performance, e.g., the accuracy tends to go down after $\alpha$ exceeds 0.005 on Flickr. This is reasonable since $\alpha$ implies a prior proportion of the causally-correlated part in a particular network as discussed in Equation 9. **(b)** G-CENSOR is relatively robust to $\beta$. It's seen that G-CENSOR can improve base models' performance under various values of $\beta$. For sensitivity analysis with GraphSAGE and GIN, refers to Appendix F.

Table 3: Comparison in node classification accuracy on out-of-distribution test set. **Bold** and underline indicate the best the second best performance, respectively. Δ represents the average accuracy degradation of base GNNs on the testing set compared to the accuracy on the validation set. ↑ represents the average gain of G-CENSOR against the best SOTA GCL methods.

| | Method | CiteSeer | Computers | Cora_Full | Cora_ML | DBLP | Flickr | Photo | PubMed |
|---|---|---|---|---|---|---|---|---|---|
| | Δ | -1.21% | 4.76% | -46.81% | -5.18% | 15.80% | -12.31% | -19.19% | -0.09% |
| | ↑ | 1.08% | 0.88% | 2.89% | 0.54% | 0.80% | 2.95% | 3.05% | 0.74% |
| SAGE | BASE | 90.29±0.68 | 82.89±12.99 | 31.88±5.88 | 80.05±4.98 | 91.22±0.83 | 46.66±0.37 | 75.57±6.13 | 87.26±0.52 |
| | MVGRL | 68.41±2.60 | OOM | OOM | 51.24±6.09 | 86.90±1.43 | TLE | OOM | 87.65±0.73 |
| | GRACE | 89.77±0.48 | 91.15±1.12 | 32.34±2.88 | 83.24±3.70 | 91.67±0.52 | 46.58±0.56 | 77.65±5.59 | 87.88±0.56 |
| | GGA | 89.77±1.60 | 88.85±3.26 | 33.78±4.04 | 83.64±2.92 | 91.97±1.46 | 46.29±0.43 | 76.12±8.19 | 88.06±0.62 |
| | BGRL | 89.65±0.73 | 88.58±3.45 | 33.93±2.19 | 83.11±3.58 | 92.42±0.58 | 46.55±0.30 | 80.22±6.55 | 88.11±0.87 |
| | RGCL | 89.29±1.69 | 51.95±40.35 | 25.46±4.34 | 79.63±5.02 | 91.99±0.99 | 36.80±0.10 | 49.18±28.98 | 86.86±0.33 |
| | AUTOGCL | 89.61±0.75 | 86.44±7.83 | 32.12±1.32 | 82.27±5.09 | 91.74±0.60 | 36.69±4.84 | 74.76±3.27 | 87.02±0.58 |
| | G-CENSOR | **91.15±1.03** | **92.62±0.23** | **36.44±3.38** | **84.36±3.99** | **93.31±0.30** | **47.51±0.24** | **84.42±2.11** | **88.77±0.39** |
| GAT | BASE | 89.98±0.91 | 89.98±4.48 | 36.99±3.82 | 83.30±2.96 | 91.70±1.13 | 43.96±4.17 | 74.74±9.61 | 87.46±1.06 |
| | MVGRL | 88.45±0.96 | OOM | OOM | 81.96±3.96 | 92.07±1.04 | TLE | OOM | 87.81±0.63 |
| | GRACE | 88.93±1.00 | 93.15±1.08 | 39.87±4.66 | 84.19±2.74 | 93.14±0.32 | 47.38±0.60 | 86.14±2.50 | 87.74±0.34 |
| | GGA | 90.05±0.97 | 93.27±0.49 | **40.28±1.63** | 84.75±2.73 | 92.96±0.45 | 46.79±0.44 | 86.80±3.12 | 88.26±0.29 |
| | BGRL | 90.37±0.60 | 91.96±0.79 | 39.49±3.42 | 83.81±3.45 | 92.74±0.68 | 44.94±0.68 | 81.15±3.33 | 88.04±0.69 |
| | RGCL | 88.37±1.61 | 55.5±37.86 | 22.75±3.24 | 79.01±4.49 | 92.40±0.71 | 37.67±3.49 | 66.25±10.34 | 86.94±0.36 |
| | AUTOGCL | 89.96±0.64 | 91.97±1.02 | 37.96±3.01 | 82.67±4.03 | 92.41±0.42 | 46.20±2.46 | 85.35±3.06 | 87.44±0.48 |
| | G-CENSOR | **91.44±0.75** | **93.29±0.45** | 40.19±2.00 | **85.35±1.84** | **93.40±0.50** | 47.22±0.35 | 85.98±1.73 | **88.62±0.32** |
| GIN | BASE | 88.97±1.17 | 90.15±1.84 | 31.24±3.57 | 80.51±5.39 | 89.93±1.16 | 44.54±2.47 | 72.64±10.94 | 88.17±0.61 |
| | MVGRL | 86.85±1.82 | OOM | OOM | 69.85±6.36 | 92.02±1.24 | TLE | OOM | 87.64±0.75 |
| | GRACE | 89.57±1.02 | 92.81±0.87 | 32.82±1.03 | 82.98±3.37 | 86.43±1.42 | 38.11±7.02 | 81.99±2.22 | 87.72±0.77 |
| | GGA | 89.29±1.33 | 90.47±2.27 | 33.59±1.02 | 83.11±3.53 | 88.62±1.34 | 41.04±1.11 | 81.21±4.00 | 88.08±0.60 |
| | BGRL | 89.49±0.49 | 91.96±0.73 | 34.74±2.35 | 82.29±3.29 | 91.02±1.37 | 38.22±5.15 | 77.82±10.65 | 88.19±0.54 |
| | RGCL | 88.08±1.00 | 86.98±2.55 | 31.11±2.97 | 81.96±2.34 | 92.18±0.53 | 41.46±3.70 | 80.43±1.63 | 86.72±0.45 |
| | AUTOGCL | 87.68±1.30 | 90.48±0.93 | 28.27±5.45 | 78.89±4.57 | 88.54±0.64 | 33.57±1.67 | 77.12±13.32 | 87.19±2.11 |
| | G-CENSOR | **90.56±0.45** | **93.75±0.53** | **35.26±1.00** | 83.16±3.13 | **93.26±0.35** | **47.82±0.41** | 85.98±1.03 | **89.13±0.34** |

[1] OOM means Out Of Memory (>32GB) and TLE means Time Limit Exceeded (seconds per epoch > 1000s)

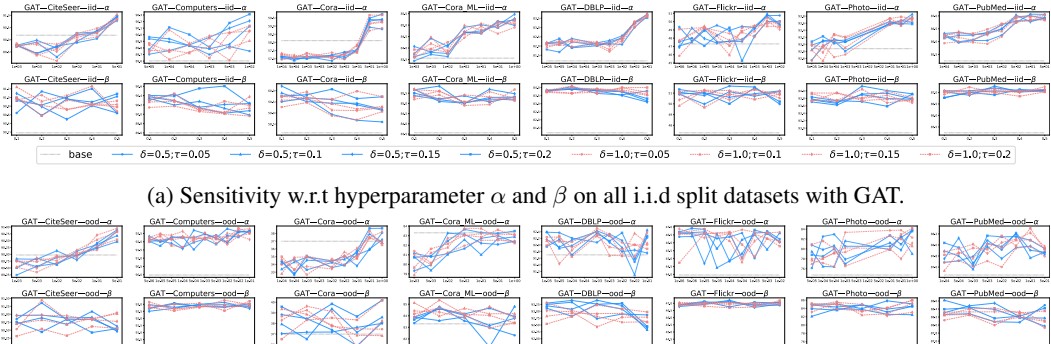

(a) Sensitivity w.r.t hyperparameter $\alpha$ and $\beta$ on all i.i.d split datasets with GAT.

(b) Sensitivity w.r.t hyperparameter $\alpha$ and $\beta$ on all o.o.d split datasets with GAT.

Figure 5: Sensitivity analysis on hyperparameters with GAT as base model.

## 6 CONCLUSION

This paper proposes a novel graph contrastive learning framework G-CENSOR, an approach leveraging task-oriented counterfactual views generation, to enhance the performance and generalizability of GNNs on node property prediction tasks without any change on the model structure and the inference flow. Through extensive experiments with in-depth analysis, we demonstrate the superiority of G-CENSOR. However, the counterfactual data synthesis can further be improved based on counterfactual inference, i.e., the three steps of abduction, action and prediction in a structural causal model (Pearl, 2009). We will explore it in the future work.

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

# Appendices

## A  EGO-GRAPH GENERATION PROCESS

### A.1  INTERPRETATIVE EXAMPLES FOR 1-HOP EGO-GRAPH

**Example 1** (For Figure 2a). Assuming this is a paper impact (y) prediction task defined on a citation graph. Generally, the type of v affects the graph structure composed of cited papers ($E^1$ with $V^1$), e.g., survey papers are more likely to be cited than general papers. Meanwhile the type of v affects the graph structure composed of references ($E^0$ with $V^0$), e.g., survey papers have more references than general papers. While being cited more ($E^1$) stably indicating higher impact (y), a GNN use all observed structures (G) is prone to learn that papers with more reference ($E^0$) tend to get higher impact (y), and this relationship can be considered as a spurious correlation.

**Example 2** (For Figure 2b). Assuming this is an image category (y) prediction task, e.g. landscape and city, defined on an image graph. In this graph, an edge exists if two images share some common properties, e.g., location, producer and object (McAuley & Leskovec, 2012). ($E^1$ with $V^1$) can be images with same objects (related to category) and ($E^0$ with $V^0$) can be images with same producers or locations. While category (y) stably indicating specific objects thus generate connections between (y) and ($E^1$), a GNN use all observed structures (G) may learn a joint probability distribution on category (y) and images sharing same producers or locations ($E^0$). This joint probability distribution can be considered as a spurious correlation.

**Example 3** (For Assumption 1). In Example 1, all highly-cited papers are necessarily considered as high-impact papers, but there can exist a high-impact paper without many references. Similarly, in Example 2, an image must have connections to those images that belong to the same category, but an landscape image can be produced by a photographer who often takes pictures of human models.

## A.2 Causal diagrams for k-hop ego-graph generation process

Let us explain a $k$-hop ego-graph generation process by a hierarchical traversal allowing revisiting a node or an edge. The causal diagrams for $k$-hop ego-graph are shown in Figure 6.

Figure 6a. In this case we assume an edge must have at least one path to the ego, where edges on the path are all causal w.r.t a task, when the edge itself is an causal edge w.r.t. the task. This ensures that the causal ego-subgraph is a connected graph. Now if the ego-graph is acyclic, there are three type of edges: (a) edges whose path(s) to the ego only include causal edges (i.e., edges indicated by the arrows in the upper row), (b) edges whose path(s) to the ego include no causal edges (i.e., edges indicated by the arrows in the bottom row), and (c) edges whose path(s) to the ego include both causal and non-causal edges (i.e., edges formed by the slanted arrows). Edges of the first type can be causally-correlated since they determine the label y, but edges of the second type can be spuriously-correlated since the association path from them to the label y exist fork (confounding) patterns joined by v. As for edges of the last type, they and label y can be also confounded by $V^{i,1}$, where $i \in \{0, 1, \ldots, k - 1\}$. If the ego-graph has cycles, which means there exists at least one edge can be included in multiple $E^{i,j}$, where $j \in \{0, 1\}$. This edge is causally-correlated to the label y if and only if $\max(j) = 1$, which indicates that this edge is part of causes of the label y, otherwise it is spuriously-correlated to the label y since it can be confounded by v like other normal edges in $E^{i,0}$.

Figure 6b. Actually high-order edges in this case has no causal relationship to the label y and they can be confounded by either v or $V^{1,1}$.

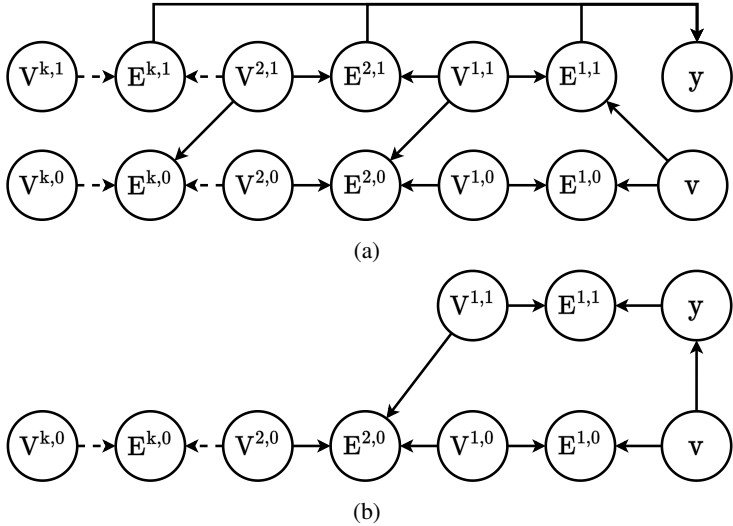

Figure 6: Causal diagrams (Pearl, 2009) for the $k$-hop ($k > 1$) ego-graph generation process. A solid arrow from a to b represents a causal relationship from a to b. The two dashed line with text GNN mean that we usually use the whole observed graph (G) to predict task label y.

## B Base Models

**GraphSAGE (abbreviated as SAGE).** Hamilton et al. Hamilton et al. (2017) proposed three variants and we utilize the simple but popular version, SAGE-mean, which directly aggregates neighbors by averaging the embeddings of them. In particular, the strategy of SAGE-mean is defined as follows:

$$\boldsymbol{h}_v^k \leftarrow \sigma \left( \boldsymbol{W}^k \cdot \left( \boldsymbol{h}_v^{k-1} \, \| \, \mathrm{Mean} \left( \{\boldsymbol{h}_v^{k-1}\} \cup \{\boldsymbol{h}_u^{k-1}, \forall u \in \mathcal{N}(v)\} \right) \right) \right),$$

where $\boldsymbol{W}^k$ is the linear transformation weight matrix in the $k$-th layer, $\boldsymbol{h}_v^k$ is the embedding of node $v$ in the $k$-th layer and $\mathcal{N}(v)$ is the neighbors of node $v$.

**GAT.** A graph attention network model employs a multi-head attention mechanism on the aggregation of neighbors' features, which enables specifying different weights to different neigh-

bors Veličković et al. (2018). The aggregation strategy is defined as:

$$
\boldsymbol{h}_v^k \leftarrow \Big\|_{m=1}^{M} \sigma^1 \left( \sum_{\forall u \in \{v\} \cup \mathcal{N}(v)} \alpha_{vu}^{mk} \boldsymbol{W}^{mk} \boldsymbol{h}_u^{k-1} \right),
$$

where $\sigma^1$ is the ELU Clevert et al. (2016) function, $\boldsymbol{W}^{mk}$ is the linear transformation weight matrix of $m$-th head in the $k$-th layer, and

$$
\alpha_{vu}^{mk} = \frac{\exp \left( \sigma^2 \left( \boldsymbol{a}^{mk} \left[ \boldsymbol{W}^{mk} \boldsymbol{h}_v^{k-1} \| \boldsymbol{W}^{mk} \boldsymbol{h}_u^{k-1} \right] \right) \right)}{\sum_{\forall i \in \{v\} \cup N(v)} \exp \left( \sigma^2 \left( \boldsymbol{a}^{mk} \left[ \boldsymbol{W}^{mk} \boldsymbol{h}_v^{k-1} \| \boldsymbol{W}^{mk} \boldsymbol{h}_i^{k-1} \right] \right) \right)},
$$

where $\sigma^2$ is the LeakyReLU Maas et al. (2013) function and $\boldsymbol{a}^{mk}$ is the weight vector of the $m$-th head in the $k$-th layer.

**GIN.** Graph Isomorphism Network, abbreviated as GIN, captures graph structure differences by summing neighbors' features, which provably maps any two graphs that the Weisfeiler-Lehman test Leman & Weisfeiler (1968) of isomorphism decides as non-isomorphic, to different embeddings Xu et al. (2019). The updating strategy is shown below

$$
\boldsymbol{h}_v^k \leftarrow \mathrm{MLP}^k \left( \left(1 + \epsilon^k \right) \boldsymbol{h}_v^{k-1} + \sum_{\forall u \in N(v)} \boldsymbol{h}_u^{k-1} \right),
$$

where $\mathrm{MLP}^k$ is a multi-layer perceptrons Hornik et al. (1989) for the $k$-th layer, and $\epsilon^k$ can be a learnable parameter or a fixed scalar in the $k$-th layer.

## C REPARAMETERIZATION TRICK

Following (Maddison et al., 2017), denoting $\frac{1}{m} \sum_{i=1}^{m} \mathrm{MLP}_i \left( \boldsymbol{e}_{u\varsigma|G_v}; \Lambda_i \right)$ in Equation (4) by $s_{u\varsigma|G_v}$, in the training stage, the probability of the edge $(u, \varsigma)$ being part of $E_v^+$ is given by

$$
\theta_{u\varsigma|G_v}^{train} = S \left( \frac{\log(\epsilon) - \log(1 - \epsilon) + s_{u\varsigma|G_v}}{\lambda} \right),
$$

where $S(x) = \frac{1}{1+e^{-x}}$ and $\epsilon \sim \mathcal{U}(0, 1)$ is an independent random variable that obeys a standard uniform distribution. $\lambda$ is the temperature parameter to control the approximation. When $\lambda \rightarrow 0$, $\theta_{u\varsigma|G_v}^{train}$ is binarized with

$$
\lim_{\tau \rightarrow 0} P \left( \theta_{u\varsigma|G_v}^{train} = 1 \right) = \frac{\exp \left( s_{u\varsigma|G_v} \right)}{1 + \exp \left( s_{u\varsigma|G_v} \right)}.
$$

## D REAL RUNTIME EVALUATION

The real runtime of a model on a dataset is calculated by averaging the minimum one-epoch runtime of the three base GNNs at all settings on the dataset (As shown in Table 4). Note that all the experiments were conducted on the same platform described in Appendix E.3. **It's seen that G-CENSOR actually achieves a very fast speed compared to other baselines.** This should be attributed to the two reasons mentioned before, i.e., simple edge probability estimator and no need to contrast in-batch samples.

## E EXPERIMENTAL SETUP

### E.1 DATASETS

In the five citation networks, i.e., Cora_Full, Cora_ML, CiteSeer, DBLP and PubMed, nodes represent papers and edges represent citation links. Given paper text as bag-of-words node features, the task is to predict the topic of a paper. In the two product networks, i.e., Computers and Photo,

Table 4: Real runtime (seconds per epoch) of all the methods on all the datasets.

| Dataset | BASE | MVGRL | GRACE | GCA | BGRL | RGCL | AUTOGCL | G-CENSOR |
|---|---|---|---|---|---|---|---|---|
| CiteSeer | 2.04±0.09 | 2.50±0.41 | 1.75±0.10 | 3.05±0.16 | 2.80±0.57 | 1.71±0.07 | 2.03±0.13 | 2.61±0.52 |
| Computers | 5.02±0.11 | OOM | 13.13±2.64 | 10.35±0.19 | 12.16±0.23 | 53.4±12.15 | 8.16±0.49 | 9.95±0.22 |
| Cora_Full | 8.41±0.11 | OOM | 29.43±6.99 | 22.80±4.71 | 31.25±11.73 | 31.6±0.47 | 12.6±1.17 | 12.31±0.94 |
| Cora_ML | 2.10±0.17 | 3.25±0.12 | 2.60±0.46 | 2.87±0.74 | 2.71±0.16 | 2.12±0.06 | 2.05±0.07 | 2.36±0.54 |
| DBLP | 3.66±0.17 | 23.36±0.53 | 9.67±0.70 | 16.17±5.19 | 9.22±0.55 | 11.15±0.57 | 5.04±0.15 | 4.92±0.85 |
| Flickr | 9.81±0.59 | >1000 | 28.35±3.13 | 46.36±15.96 | 59.72±36.26 | 110.66±15.47 | 17.91±1.00 | 14.93±2.43 |
| Photo | 3.06±0.06 | OOM | 5.64±1.28 | 4.45±0.07 | 5.53±0.09 | 15.35±2.49 | 4.01±0.11 | 4.42±0.18 |
| PubMed | 3.13±0.15 | 27.12±1.32 | 8.20±1.23 | 7.37±1.59 | 8.10±1.22 | 10.02±0.19 | 4.49±0.26 | 4.56±0.42 |

nodes represent products and edges represent that two goods are frequently bought together. Given product reviews as bag-of-words node features, the task is to map goods to their respective product category. In the image network, i.e., Flickr, nodes represent images and edges represent that two images share some common properties (e.g., same geographic location and comments by the same user, etc.). Given bag-of-word representation of the images as node features, the task is to predict the type of an image.

Table 5: Datasets statistics

| Dataset | #nodes | #edges | #features | #classes |
|---|---|---|---|---|
| Cora_Full | 19,793 | 126,842 | 8,710 | 70 |
| Cora_ML | 2,995 | 16,316 | 2,879 | 7 |
| CiteSeer | 4,230 | 10,674 | 602 | 6 |
| DBLP | 17,716 | 105,734 | 1,639 | 4 |
| PubMed | 19,717 | 88,648 | 500 | 3 |
| Computers | 13,752 | 491,722 | 767 | 10 |
| Photo | 7,650 | 238,162 | 745 | 8 |
| Flickr | 89,250 | 899,756 | 500 | 7 |

## E.2 BASELINES

Base models, i.e., **SAGE**, **GAT** and **GIN**, refer to Appendix B.

GCL methods with uniform or adaptive data augmentation:

1. **MVGRL(Hassani & Ahmadi, 2020):** Multi-View Graph Representation Learning, an approach contrasting encodings from first-order neighbors and a general graph diffusion and also contrasting node and graph encodings across views.

2. **Grace(Zhu et al., 2020):** GRAph Contrastive rEpresentation learning, an approach generating two graph views by corruption and learn node representation by maximizing the agreement of node representations in these two views.

3. **GCA(Zhu et al., 2021):** Graph Contrastive representation learning with Adaptive augmentation, an approach designing augmentation scheme based on node centrality measures to highlight important connective structures.

4. **BGRL(Thakoor et al., 2021):** Bootstrapped Graph Latents, an graph representation learning method that learns by predicting alternative augmentations of the input. BGRL uses only simple augmentations and alleviates the need for contrasting with negative examples, and is thus scalable by design.

GCL methods with learnable data augmentation:

1. **RGCL(Li et al., 2022):** Rationale-aware Graph Contrastive Learning, an unsupervised approach using a rationale generator to reveal salient structures about graph instance-discrimination as the rationale, and then creating rationale-aware views for contrastive learning. Note that this method, designed for graph property prediction tasks, integrates the views generation module and the inference flow of the predictor. Therefore, we regard node property prediction tasks as ego-graph property prediction tasks to adapt to this method.

2. **AutoGCL(Yin et al., 2022):** Automated Graph Contrastive Learning, an approach employing a set of learnable graph view generators orchestrated by an auto augmentation strategy, where every graph view generator learns a probability distribution of graphs conditioned by the input. This method is proposed for graph property prediction tasks. However, it can be directly transferred to node property tasks since its views generator and task predictor are separates.

Note that while our work can be easily enhanced by considering node feature transformation in views generation, we focus on structure transformation in this work, thus feature transformation is disabled in all models including G-CENSOR.

### E.3 IMPLEMENTATION DETAILS

Experiments were conducted on a Ubuntu 18.04 server with one Nvidia Tesla V100-32G GPU. And the code was implemented using python 3.8 with PyG 2.0.4 and Pytorch 1.11 that used CUDA version 11.3. For all datasets, the number of sampled neighbors was set to 64. The batch size was set to 64 for all models and an AdamW optimizer (Loshchilov & Hutter, 2019) with learning rate 0.01 was used to train all models.

For a fair comparison, the number of layers of base GNNs was set to 2 for all baselines and all contrastive baselines were used as an auxiliary task (Xie et al., 2022). The weight of the contrastive loss was searched from 0.1 to 0.9. Moreover, for all baselines with hyperparameters of edge drop probabilities and temperature, we searched edge drop probabilities over [0.1, 0.2, 0.3, 0.4] and searched temperatures over [0.1, 0.2], unless the original paper reported the best choices on the datasets. As for G-CENSOR, $m$ in Equation 4 was simply set to 4, $\alpha$ in Equation 9 was searched from 1e-5 to 1e-1, $\delta_{margin}$ in Equation 7 was searched over [0.5, 0.1], $\tau$ in Equation 10 was searched over [0.05, 0.1, 0.15, 0.2], and $\beta$ in Equation 11 was searched from 0.1 to 0.5.

## F SENSITIVITY ANALYSIS

This section displays the sensitivity analysis on hyperparameters for GraphSAGE and GIN.

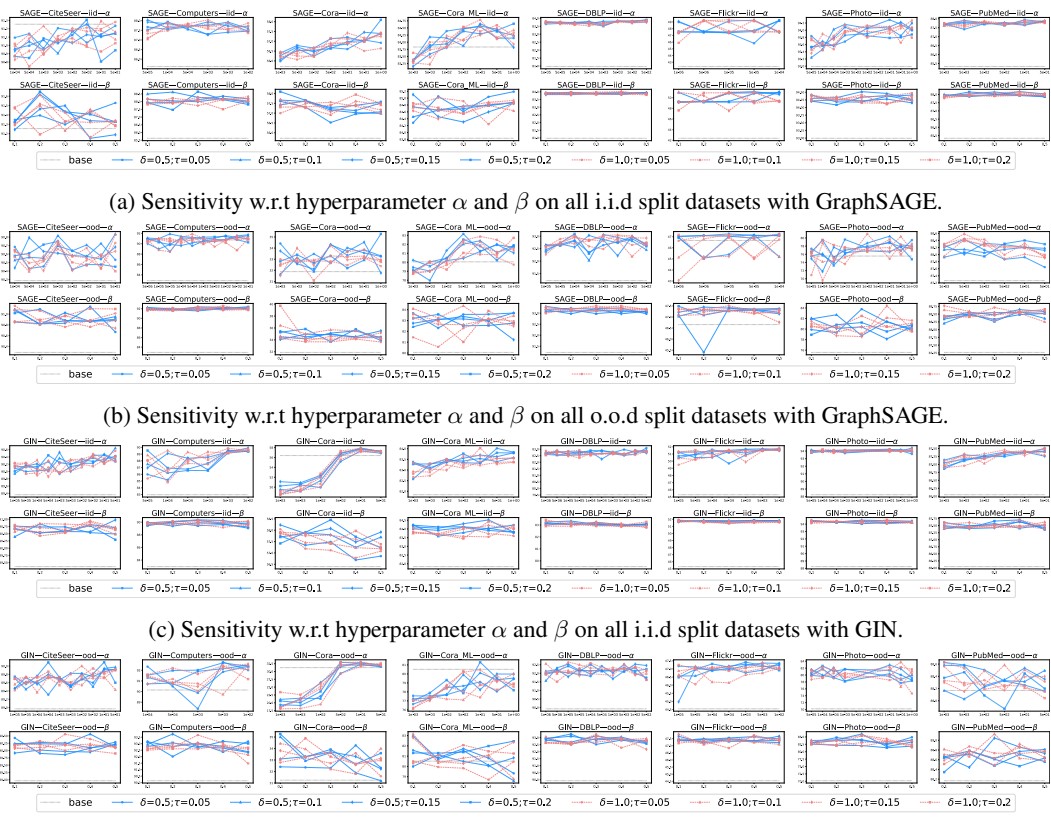

(a) Sensitivity w.r.t hyperparameter $\alpha$ and $\beta$ on all i.i.d split datasets with GraphSAGE.

(b) Sensitivity w.r.t hyperparameter $\alpha$ and $\beta$ on all o.o.d split datasets with GraphSAGE.

(c) Sensitivity w.r.t hyperparameter $\alpha$ and $\beta$ on all i.i.d split datasets with GIN.

(d) Sensitivity w.r.t hyperparameter $\alpha$ and $\beta$ on all o.o.d split datasets with GIN.

Figure 7: Sensitivity analysis on hyperparameters with GraphSAGE and GIN.

