# OpenReview forum: "G-Censor: Graph Contrastive Learning with Task-Oriented Counterfactual Views"
_ICLR.cc/2023/Conference — Submitted to ICLR 2023_

### Official Review · Reviewer_tYMD · 2022-10-19

**Confidence:** 5
**Correctness:** 3
**Technical Novelty And Significance:** 2
**Empirical Novelty And Significance:** 2
**Recommendation:** 3

**Clarity, Quality, Novelty And Reproducibility:**

Clarity: the idea and method are well clarified in this paper.

Quality: counterfactual perspective is claimed to be the core contribution of this paper, but I cannot see the intrinsic difference between the counterfactual samples (edge-wise) in this paper and the rationale-aware augmented views (node-wise) in RGCL [1].

Novelty: limited tech novelty. DIR[2], AD-GCL[3] and RGCL[1] leverage an auxiliary model to identify crucial nodes/edges to construct augmented views. AD-GCL[3] adopts Gumbel reparameterization trick to enable the gradient propagation. The workflow of the parameterizing the sampling probability and constructing positive/negative views is very similar to RGCL[1]. The pipeline of G-CENSOR is more like replanting aforementioned techs into supervised node-level task.

Reproducibility: the authors will release the codes after the review, thus I cannot comment on the reproducibility now.
[1]	Li, Sihang, et al. "Let invariant rationale discovery inspire graph contrastive learning." International Conference on Machine Learning. PMLR, 2022.
[2]	Wu, Yingxin, et al. "Discovering Invariant Rationales for Graph Neural Networks." International Conference on Learning Representations. 2021.
[3]	Suresh, Susheel, et al. "Adversarial graph augmentation to improve graph contrastive learning." Advances in Neural Information Processing Systems 34 (2021): 15920-15933.


**Strength And Weaknesses:**

Strength
(1) Overall, this paper is well-written and the idea is easy to follow.
(2) Far as I know, this paper is the first to study the construction of semantics-preserved augmented views for GCL on node-level tasks.
(3) The results showcase the effectiveness of G-CENSOR.

Weakness:
(1) What is the difference between counterfactual view and augmented view? Please see following Quality for detailed comments.
(2) Limited technical novelty. Please see following Novelty for detailed comments.
(3) Lack of empirical validation of CTM. CTM is supposed to discover “edges causally related to the task label”. However, no empirical or theoretical evidence is presented to validate this statement. Concretely speaking, I would prefer to see an ego-graph and CTM’s estimation of each edge’s sampling probability to validate its ability of revealing “edges causally related to the task label”.
(4) Miscellaneous minor issues. Several misspelling (a -> an Line 8 Para 2 Page 1) and citation format problem (\cite or \citep command).


**Summary Of The Paper:**

The authors proposed a supervised framework -- G-CENSOR, which utilizes an auxiliary model (i.e., CTM) to estimate the semantic importance of each edge in an ego-graph and transfer them into sampling probabilities to construct positive and negative views. Further, the model is optimized by minimizing three sub-losses: (1) prediction loss i.e., conventional supervised loss, (2) counterfactual loss, which gives the positive view the same supervision signal as the original sample and enlarges the rank gap between positive and negative views, and (3) contrastive loss, an infoNCE style loss with comparison among anchor sample and its augmented views. The experiments results show the effectiveness of G-CENSOR on network datasets.

**Summary Of The Review:**

The authors proposed G-CENSOR, a model-agnostic supervised framework on node-level task with the assistance of contrastive learning. And extensive experiment show the effectiveness of it. However, the tech novelty is rather limited and it lacks empirical or theoretical results to demonstrate the effectiveness of the core module -- CTM.

---

### Official Review · Reviewer_DcTY · 2022-10-23

**Confidence:** 4
**Correctness:** 3
**Technical Novelty And Significance:** 3
**Empirical Novelty And Significance:** 3
**Recommendation:** 5

**Clarity, Quality, Novelty And Reproducibility:**

The paper is presented in a clear way, the method is novel, and and the authors mentioned that the code will be available after publication.

**Strength And Weaknesses:**

# Strength
- The studied problem of optimal task-oriented contrastive view generation is important.
- The paper is writing in a clear way and easy to follow.
- The authors provide extensive experiments on benchmark datasets.

# Weakness
- Assumption 1 assumes that the prediction loss based on $\phi(G^-)$ should be higher than $\phi(G^+)$. However, as $G^-$ often has spurious correlations with the label y, the prediction based on $\phi(G^-)$ is possible to be even more accurate than $\phi(G^+)$. How can we verify this assumption?
- Similarly, in the discussion about the difference between counterfactual explanation and this work, the authors mentioned that compared with directing maximizing $\mathcal{L}_{pred}(\phi^-)$, the rank loss used in Eq. (7) can mitigate the spurious correlation problem. However, I feel this claim is not convincing enough, and hope the authors can further justify it.
- The counterfactual transformation module estimates the probability of each edge to be part of the optimal positive view. I wonder whether an edge-level prediction is good enough for view generation,  especially when more higher-level substructure (e.g., triangle) are involved.
- In Definition 2, I suggest the authors to double check the notation. It seems $\bar{G}’$ there should be $G’$.


**Summary Of The Paper:**

This paper studies the problem of generating optimal task-oriented contrastive views in graph contrastive learning (GCL). Existing GCL methods find this problem challenging when the task-related semantics are incomplete in positive/negative views.  For this problem, the authors propose G-CENSOR, which is a model-agnostic method to generate optimal task-oriented counterfactual views for graphs and GNNs in node property prediction. Experiments are conducted on eight real-world datasets, and the results show that the proposed method outperforms state-of-the-art GCL methods in task performance and generalizability.

**Summary Of The Review:**

Overall, the studied problem of this paper is important. The proposed method is novel and sound generally. The experiments include extensive evaluations on benchmark datasets, and the results look encouraging. But my major concern is that the key assumption and corresponding technique design need further justification. That is the main reason I give a weak reject now.

---

### Official Review · Reviewer_iyAN · 2022-10-24

**Confidence:** 4
**Clarity, Quality, Novelty And Reproducibility:** 1. Starting from a hypothesis, this p…
**Correctness:** 3
**Technical Novelty And Significance:** 3
**Empirical Novelty And Significance:** 2
**Recommendation:** 5

**Strength And Weaknesses:**

**Strength**
1. The problem formulation of graph contrastive learning is quite detailed. And there is also a discussion of the causality of the proposed method.
2. The idea of automatic generation of negative views is practical to get rid of the performance-memory trade-off inherent.
3. The explanation of each part of the model is clear and the core idea is easy to follow.

**Weakness**
1. The way in which negative views are generated is sloppy and lacks detailed discussion. More ablation experiments may help explain the trade-off.
2. In the absence of released code, the experiment details in the task shift of prior methods from graph classification to node classification need to be further explained due to significant discrepancies in results.
3. The figures and captions in this article are unclear with rough details. Task B in Figure 1 lacks an explanation of how the prediction works. The estimation in Figure (b) is not intuitive.

**Summary Of The Paper:**

To answer a pre-posed question about the contrastive views of graph representation learning, this paper studies task-oriented contrastive learning from a counterfactual perspective in node property prediction tasks. A model-agnostic framework, G-Censor was proposed to generate both positive and negative views in graph contrastive learning, the performance of which is evaluated on real-world datasets. Compared with prior methods, the model proposed in this paper has the characteristic to generate learnable views, especially negative views in specific tasks.

**Summary Of The Review:**

This paper has a novel way of thinking and perspective on graph contrastive learning in specific tasks, and the formulation of the problem is also very detailed. However, the novelty is limited, and the experiment results are not that convincing compared with the previous work. Hence, I am leaning on the negative side.

---

### Official Review · Reviewer_EBjy · 2022-10-25

**Confidence:** 4
**Correctness:** 2
**Technical Novelty And Significance:** 2
**Empirical Novelty And Significance:** Not applicable
**Recommendation:** 3

**Clarity, Quality, Novelty And Reproducibility:**

Clarity: can be improved. These are several typos and redundant descriptions that hurts the readability (e.g., the two statements after “where” in Eqs 6,7 are redundant and confusing).

Quality and Novelty: The idea of CTM is very similar to those from ADGCL and RGCL.

Reproducibility: Poor. The settings for datasets and baselines lack of clear clarification and the reported performance does not align with the existing literatures.


**Strength And Weaknesses:**

S1:
The idea of improving GNNs’ performance on node classification tasks by generating counterfactual examples is an interesting and well-motivated.

W1:
It seems like representation of an edge (i.e., Section 4.2) involves some engineering tricks, so ablation studies are needed upon each of these tricks to demonstrate that the actual performance improvements are most from the proposed high-level philosophies.

W2:
The high-level idea of the counterfactual transformation module (CTM) is very similar to the augmentation module proposed in ADGCL [1], which may give the readers a feeling that the proposed method re-packages the idea in ADGCL with the concepts of counterfactual learning. I think the authors should discuss [1] in this paper.

W3:
The proposed framework involves many sub-modules and objectives, which makes it hard to judge what are the core contribution of the proposed method. I would suggest the authors to conduct extensive ablation studies to demonstrate and justify their design choices.

W4:
[2] also studied counterfactual data augmentation on graphs. The authors should also include discussion of it.

W5:
The experimental setting is not clearly described. According to my own implementation based on DGL, GIN underperforms badly on node-level tasks (e.g., almost 10% worse than vanilla GNNs on benchmarks with public splits), which does not align with the results from this paper. Besides, RGCL and AutoGCL are targeted for the graph classification task and the authors does not specify how they transfer these framework into the settings for the node classification.

[1] Adversarial graph augmentation to improve graph contrastive learning. NeurIPS 21. \
[2] Learning from counterfactual links for link prediction. ICML 22.




**Summary Of The Paper:**

This paper proposes a supervised learning framework that utilizes the philosophy of counterfactual learning to enhance the node classification performance. Specifically, this paper utilizes a counterfactual transformation module to construct sub-ego-graphs that are positively correlated to the label information, and explores the complements of these sub-ego-graphs as negative counterparts for contrastive learning, which are negatively correlated with the label information (or not as positively correlated as the former according to the formulation in this paper). The learning framework explores five objectives to simultaneously fulfill the purposes of supervised learning, counterfactual ego-graph construction, and sparsity guarantees of the ego-graphs. The proposed framework improves the performance of the node classification tasks when applied to vanilla GNN backbones.


**Summary Of The Review:**

Utilizing the concept from counterfactual learning into the graph machine learning field is interesting and well-motivated. However, several formulations and implementations seem to come from the existing works. Besides, ablation studies on multiple design choices are missing, which made it difficult to judge the research contribution of this work.

Therefore, I think this paper requires substantial revision, and I recommend rejection.


===

post rebuttal: I appreciate the detailed response. After reading all the reviews and responses, I decide to keep my score.

---

### Official Review · Reviewer_VhaN · 2022-10-26

**Confidence:** 5
**Correctness:** 3
**Technical Novelty And Significance:** 3
**Empirical Novelty And Significance:** 2
**Recommendation:** 3

**Clarity, Quality, Novelty And Reproducibility:**

Overall, the clarity is good despite several grammar mistakes. I in person believe the technical part of this work is correct. However, as mentioned in the above question, the novelty feels limited to me.

**Strength And Weaknesses:**

Strengths:
+ It is a more principled way to construct contrastive views based on counterfactual inference.
+ The idea is clear and easy to follow.

Weaknesses:
- Several important baselines should be compared:
  1. Regarding causal graph contrastive learning: Generating Counterfactual Hard Negative Samples for Graph Contrastive Learning, arXiv 2207.00148
  2. Regarding learnable view generation: Adversarial Graph Augmentation to Improve Graph Contrastive Learning, NeurIPS 2021
  3. Regarding task-oriented graph augmentation for contrastive learning: Augmentations in Graph Contrastive Learning: Current Methodological Flaws & Towards Better Practices, WWW 2021
- Evaluation is not fair as the training stage considers downstream task information. It should be noted that all considered baselines have separated training and test stages. More specifically, their models are trained with unsupervised contrastive objectives at first and then evaluated with another prediction head. In this work, the training stage considers supervision signals, which renders performance comparison unfair. I strongly encourage authors to consider supervised contrastive learning setting (Supervised Contrastive Learning, NeurIPS 2020) for all baselines.
- Performance improvements are not significant. I note the authors claim their work does not contrast in-batch negative samples, which improves memory efficiency. I would like to see more experiments on large-scale datasets to support this claim, e.g., ogbn-arXiv.
- The setting for the generalizability experiment is not clear to me. In particular, why clustering size is set to 2 is kind of tricky. Other common ways of splitting datasets can be considered, for example scaffold split in molecular graphs.
- The authors are encouraged to elaborate on the choice of graph encoders, as in Assumption 1 the graph encoders should be a sufficient encoder. What kind of encoders are considered as sufficient encoders? It feels to me that GIN will result in information loss when applying to node-level tasks.

Minor points:
- Page 1, a appropriate positive view -> an appropriate positive view
- Tables 2 and 3, GGA -> GCA
- Definitions 1 and 2, "causally correlated" should be specified in mathematical form to avoid ambiguity.

**Summary Of The Paper:**

This paper presents graph contrastive learning approach with counterfactual views. Specifically, the authors propose a counterfactual inference objective to generate task-oriented causal views that will be helpful in determining positive and negative samples. Experimental results show superiority over existing graph contrastive learning approaches.

**Summary Of The Review:**

As a summary, this paper presents an interesting approach to improve graph contrastive learning by constructing causally correlated positive and negative views. However, it seems that the empirical evaluation is weak and several claims need to be further elaborated or supported by more experimental results. Therefore at this moment I recommend rejection.

---

### Decision · Program_Chairs · 2023-01-20

**Decision:**

Reject

**Justification For Why Not Higher Score:**

This is a good first effort but the causal theory behind the method needs better formalization.

**Justification For Why Not Lower Score:**

N/A

**Metareview: Summary, Strengths And Weaknesses:**

This paper studies the task of generating optimal task-oriented contrastive views in graph contrastive learning (GCL). The goal is to improve node classification performance. Current GCL methods have difficulties there are incomplete positive & negative views. The work uses a counterfactual transformation module to construct induced ego-networks that are positively correlated to the label information, and explores the complement of these ego-nets as negative counterparts for contrastive learning, which are negatively correlated with the label information.

After reading the reviews, rebuttal, and paper, it feels like a good idea that needs more work. Informal definitions such as "causally correlated" and "counterfactual graph" need to be formalized. The work also needs a clear structural causal model and a clear counterfactual operation (using Imbens or Pearl notation) and learning procedure. A good example of a paper that did this well is https://arxiv.org/pdf/2201.12872.pdf. Start with a causal graph model (which induces a distribution over graphs and makes clear what the counterfactual do() operator is acting on). Then define the concept of counterfactual ego-nets with respect to this causal model.